# Probabilistic ODE Solvers with Runge-Kutta Means

**Michael Schober**
MPI for Intelligent Systems
Tübingen, Germany
mschober@tue.mpg.de

**David Duvenaud**
Department of Engineering
Cambridge University
dkd23@cam.ac.uk

**Philipp Hennig**
MPI for Intelligent Systems
Tübingen, Germany
phennig@tue.mpg.de

## Abstract

Runge-Kutta methods are the classic family of solvers for ordinary differential equations (ODEs), and the basis for the state of the art. Like most numerical methods, they return point estimates. We construct a family of *probabilistic* numerical methods that instead return a Gauss-Markov process defining a probability distribution over the ODE solution. In contrast to prior work, we construct this family such that posterior means match the outputs of the Runge-Kutta family exactly, thus inheriting their proven good properties. Remaining degrees of freedom not identified by the match to Runge-Kutta are chosen such that the posterior probability measure fits the observed structure of the ODE. Our results shed light on the structure of Runge-Kutta solvers from a new direction, provide a richer, probabilistic output, have low computational cost, and raise new research questions.

## 1 Introduction

Differential equations are a basic feature of dynamical systems. Hence, researchers in machine learning have repeatedly been interested in both the problem of inferring an ODE description from observed trajectories of a dynamical system [1, 2, 3, 4], and its dual, inferring a solution (a trajectory) for an ODE initial value problem (IVP) [5, 6, 7, 8]. Here we address the latter, classic numerical problem. Runge-Kutta (RK) methods [9, 10] are standard tools for this purpose. Over more than a century, these algorithms have matured into a very well-understood, efficient framework [11].

As recently pointed out by Hennig and Hauberg [6], since Runge-Kutta methods are linear extrapolation methods, their structure can be emulated by Gaussian process (GP) regression algorithms. Such an algorithm was envisioned by Skilling in 1991 [5], and the idea has recently attracted both theoretical [8] and practical [6, 7] interest. By returning a posterior probability measure over the solution of the ODE problem, instead of a point estimate, Gaussian process solvers extend the functionality of RK solvers in ways that are particularly interesting for machine learning. Solution candidates can be drawn from the posterior and marginalized [7]. This can allow probabilistic solvers to stop earlier, and to deal (approximately) with probabilistically uncertain inputs and problem definitions [6]. However, current GP ODE solvers do not share the good theoretical convergence properties of Runge-Kutta methods. Specifically, they do not have high polynomial order, explained below.

We construct GP ODE solvers whose posterior mean functions *exactly* match those of the RK families of first, second and third order. This yields a probabilistic numerical method which combines the strengths of Runge-Kutta methods with the additional functionality of GP ODE solvers. It also provides a new interpretation of the classic algorithms, raising new conceptual questions.

While our algorithm could be seen as a "Bayesian" version of the Runge-Kutta framework, a philosophically less loaded interpretation is that, where Runge-Kutta methods fit a single curve (a point estimate) to an IVP, our algorithm fits a probability distribution over such potential solutions, such that the mean of this distribution matches the Runge-Kutta estimate exactly. We find a family of models in the space of Gaussian process linear extrapolation methods with this property, and select a member of this family (fix the remaining degrees of freedom) through statistical estimation.

| $p = 1$ | $p = 2$ | $p = 3$ |
|---|---|---|

$$
\begin{array}{c|c}
0 & 0 \\
\hline
 & 1
\end{array}
\qquad
\begin{array}{c|cc}
0 & 0 & \\
\alpha & \alpha & 0 \\
\hline
 & \left(1 - \tfrac{1}{2\alpha}\right) & \tfrac{1}{2\alpha}
\end{array}
\qquad
\begin{array}{c|ccc}
0 & 0 & & \\
u & u & 0 & \\
v & v - \frac{v(v-u)}{u(2-3u)} & \frac{v(v-u)}{u(2-3u)} & 0 \\
\hline
 & 1 - \frac{2-3v}{6u(u-v)} - \frac{2-3u}{6v(v-u)} & \frac{2-3v}{6u(u-v)} & \frac{2-3u}{6v(v-u)}
\end{array}
$$

Table 1: All consistent Runge-Kutta methods of order $p \le 3$ and number of stages $s = p$ (see [11]).

## 2 Background

An ODE *Initial Value Problem (IVP)* is to find a function $x(t) : \mathbb{R} \to \mathbb{R}^N$ such that the ordinary differential equation $\dot{x} = f(x,t)$ (where $\dot{x} = \partial x / \partial t$) holds for all $t \in T = [t_0, t_H]$, and $x(t_0) = x_0$. We assume that a unique solution exists. To keep notation simple, we will treat $x$ as scalar-valued; the multivariate extension is straightforward (it involves $N$ separate GP models, explained in supp.).

Runge-Kutta methods[1] [9, 10] are carefully designed linear extrapolation methods operating on small contiguous subintervals $[t_n, t_n + h] \subset T$ of length $h$. Assume for the moment that $n = 0$. Within $[t_0, t_0 + h]$, an RK method of *stage $s$* collects evaluations $y_i = f(\hat{x}_i, t_0 + hc_i)$ at $s$ recursively defined input locations, $i = 1, \dots, s$, where $\hat{x}_i$ is constructed *linearly* from the previously-evaluated $y_{j<i}$ as

$$
\hat{x}_i = x_0 + h \sum_{j=1}^{i-1} w_{ij} y_j, \tag{1}
$$

then returns a single prediction for the solution of the IVP at $t_0 + h$, as $\hat{x}(t_0 + h) = x_0 + h \sum_{i=1}^{s} b_i y_i$ (modern variants can also construct non-probabilistic error estimates, e.g. by combining the same observations into two different RK predictions [12]). In compact form,

$$
y_i = f\left(x_0 + h \sum_{j=1}^{i-1} w_{ij} y_j, \; t_0 + hc_i\right), \quad i = 1, \dots, s, \qquad \hat{x}(t_0 + h) = x_0 + h \sum_{i=1}^{s} b_i y_i. \tag{2}
$$

$\hat{x}(t_0 + h)$ is then taken as the initial value for $t_1 = t_0 + h$ and the process is repeated until $t_n + h \ge t_H$.

A Runge-Kutta method is thus identified by a lower-triangular matrix $\boldsymbol{W} = \{w_{ij}\}$, and vectors $\boldsymbol{c} = [c_1, \dots, c_s]$, $\boldsymbol{b} = [b_1, \dots, b_s]$, often presented compactly in a *Butcher tableau* [13]:

$$
\begin{array}{c|ccccc}
c_1 & 0 & & & & \\
c_2 & w_{21} & 0 & & & \\
c_3 & w_{31} & w_{32} & 0 & & \\
\vdots & \vdots & \vdots & \ddots & \ddots & \\
c_s & w_{s1} & w_{s2} & \cdots & w_{s,s-1} & 0 \\
\hline
 & b_1 & b_2 & \cdots & b_{s-1} & b_s
\end{array}
$$

As Hennig and Hauberg [6] recently pointed out, the linear structure of the extrapolation steps in Runge-Kutta methods means that their algorithmic structure, the Butcher tableau, can be constructed naturally from a Gaussian process regression method over $x(t)$, where the $y_i$ are treated as "observations" of $\dot{x}(t_0 + hc_i)$ and the $\hat{x}_i$ are subsequent posterior estimates (more below). However, proper RK methods have structure that is not generally reproduced by an arbitrary Gaussian process prior on $x$: Their distinguishing property is that the approximation $\hat{x}$ and the Taylor series of the true solution coincide at $t_0 + h$ up to the $p$-th term—their numerical error is bounded by $\|x(t_0 + h) - \hat{x}(t_0 + h)\| \le Kh^{p+1}$ for some constant $K$ (higher orders are better, because $h$ is assumed to be small). The method is then said to be *of order $p$* [11]. A method is *consistent*, if it is of order $p = s$. This is only possible for $p < 5$ [14, 15]. There are no methods of order $p > s$. High order is a strong desideratum for ODE solvers, not currently offered by Gaussian process extrapolators.

Table 1 lists all consistent methods of order $p \le 3$ where $s = p$. For $s = 1$, only *Euler's method* (linear extrapolation) is consistent. For $s = 2$, there exists a family of methods of order $p = 2$, parametrized

by a single parameter $\alpha \in (0, 1]$, where $\alpha = 1/2$ and $\alpha = 1$ mark the *midpoint rule* and *Heun's method*, respectively. For $s = 3$, third order methods are parameterized by two variables $u, v \in (0, 1]$.

*Gaussian processes (GPs)* are well-known in the NIPS community, so we omit an introduction. We will use the standard notation $\mu : \mathbb{R} \to \mathbb{R}$ for the mean function, and $k : \mathbb{R} \times \mathbb{R} \to \mathbb{R}$ for the covariance function; $k_{UV}$ for Gram matrices of kernel values $k(u_i, v_j)$, and analogous for the mean function: $\mu_T = [\mu(t_1), \ldots, \mu(t_N)]$. A GP prior $p(x) = \mathcal{GP}(x; \mu, k)$ and observations $(T, Y) = \{(t_1, y_1), \ldots, (t_s, y_s)\}$ having likelihood $\mathcal{N}(Y; x_T, \Lambda)$ give rise to a posterior $\mathcal{GP}^s(x; \mu^s, k^s)$ with

$$\mu_t^s = \mu_t + k_{tT}(k_{TT} + \Lambda)^{-1}(Y - \mu_T) \qquad \text{and} \qquad k_{uv}^s = k_{uv} - k_{uT}(k_{TT} + \Lambda)^{-1}k_{Tv}. \qquad (3)$$

GPs are closed under linear maps. In particular, the joint distribution over $x$ and its derivative is

$$p\left[\begin{pmatrix} x \\ \dot{x} \end{pmatrix}\right] = \mathcal{GP}\left[\begin{pmatrix} x \\ \dot{x} \end{pmatrix}; \begin{pmatrix} \mu \\ \mu^\partial \end{pmatrix}, \begin{pmatrix} k & k^\partial \\ \partial k & \partial k^\partial \end{pmatrix}\right] \qquad (4)$$

$$\text{with} \qquad \mu^\partial = \frac{\partial \mu(t)}{\partial t}, \quad k^\partial = \frac{\partial k(t, t')}{\partial t'}, \quad \partial k = \frac{\partial k(t, t')}{\partial t}, \quad \partial k^\partial = \frac{\partial^2 k(t, t')}{\partial t \partial t'}. \qquad (5)$$

A recursive algorithm analogous to RK methods can be constructed [5, 6] by setting the prior mean to the constant $\mu(t) = x_0$, then recursively estimating $\hat{x}_i$ in some form from the current posterior over $x$. The choice in [6] is to set $\hat{x}_i = \mu^i(t_0 + hc_i)$. "Observations" $y_i = f(\hat{x}_i, t_0 + hc_i)$ are then incorporated with likelihood $p(y_i | x) = \mathcal{N}(y_i; \dot{x}(t_0 + hc_i), \Lambda)$. This recursively gives estimates

$$\hat{x}(t_0 + hc_i) = x_0 + \sum_{j=1}^{i-1}\sum_{\ell=1}^{i-1} k^\partial(t_0 + hc_i, t_0 + hc_\ell)(\partial K^\partial + \Lambda)^{-1}_{\ell j} y_j = x_0 + h\sum_j w_{ij} y_j, \qquad (6)$$

with $\partial K^\partial{}_{ij} = \partial k^\partial(t_0 + hc_i, t_0 + hc_j)$. The final prediction is the posterior mean at this point:

$$\hat{x}(t_0 + h) = x_0 + \sum_{i=1}^{s}\sum_{j=1}^{s} k^\partial(t_0 + h, t_0 + hc_j)(\partial K^\partial + \Lambda)^{-1}_{ji} y_i = x_0 + h\sum_i b_i y_i. \qquad (7)$$

## 3    Results

The described GP ODE estimate shares the algorithmic structure of RK methods (i.e. they both use weighted sums of the constructed estimates to extrapolate). However, in RK methods, weights and evaluation positions are found by careful analysis of the Taylor series of $f$, such that low-order terms cancel. In GP ODE solvers they arise, perhaps more naturally but also with less structure, by the choice of the $c_i$ and the kernel. In previous work [6, 7], both were chosen ad hoc, with no guarantee of convergence order. In fact, as is shown in the supplements, the choices in these two works—square-exponential kernel with finite length-scale, evaluations at the predictive mean—do not even give the first order convergence of Euler's method. Below we present three specific regression models based on integrated Wiener covariance functions and specific evaluation points. Each model is the improper limit of a Gauss-Markov process, such that the posterior distribution after $s$ evaluations is a proper Gaussian process, and the posterior mean function at $t_0 + h$ coincides *exactly* with the Runge-Kutta estimate. We will call these methods, which give a probabilistic interpretation to RK methods and extend them to return probability distributions, *Gauss-Markov-Runge-Kutta (GMRK) methods*, because they are based on Gauss-Markov priors and yield Runge-Kutta predictions.

### 3.1    Design choices and desiderata for a probabilistic ODE solver

Although we are not the first to attempt constructing an ODE solver that returns a probability distribution, open questions still remain about what, exactly, the properties of such a probabilistic numerical method should be. Chkrebtii et al. [8] previously made the case that Gaussian measures are uniquely suited because solution spaces of ODEs are Banach spaces, and provided results on consistency. Above, we added the desideratum for the posterior mean to have high order, i.e. to reproduce the Runge-Kutta estimate. Below, three additional issues become apparent:

**Motivation of evaluation points**    Both Skilling [5] and Hennig and Hauberg [6] propose to put the "nodes" $\hat{x}(t_0 + hc_i)$ at the current posterior mean of the belief. We will find that this can be made

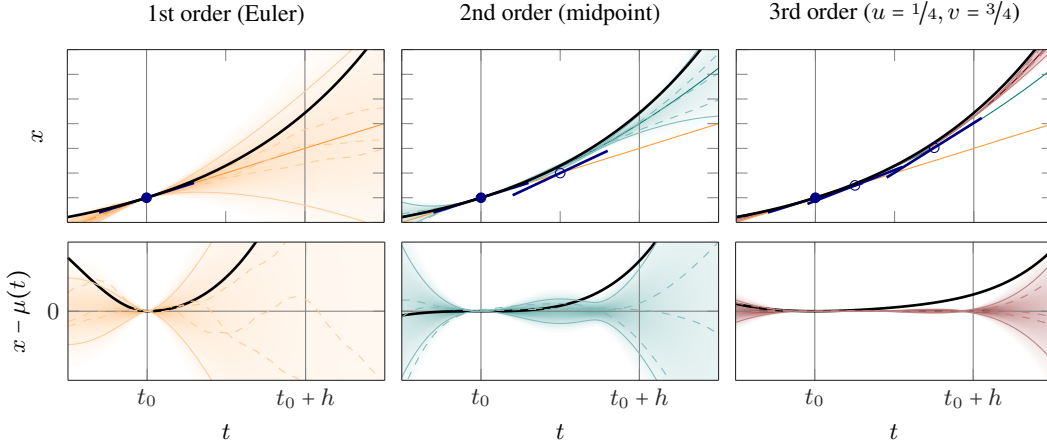

Figure 1: **Top:** Conceptual sketches. Prior mean in gray. Initial value at $t_0 = 1$ (filled blue). Gradient evaluations (empty blue circles, lines). Posterior (means) after first, second and third gradient observation in orange, green and red respectively. Samples from the final posterior as dashed lines. Since, for the second and third-order methods, only the final prediction is a proper probability distribution, for intermediate steps only mean functions are shown. True solution to (linear) ODE in black. **Bottom:** For better visibility, same data as above, minus final posterior mean.

consistent with the order requirement for the RK methods of first and second order. However, our third-order methods will be forced to use a node $\hat{x}(t_0 + hc_i)$ that, albeit lying along a function $w(t)$ in the reproducing kernel Hilbert space associated with the posterior GP covariance function, is not the mean function itself. It will remain open whether the algorithm can be amended to remove this blemish. However, as the nodes do not enter the GP regression formulation, their choice does not directly affect the probabilistic interpretation.

**Extension beyond the first extrapolation interval**    Importantly, the Runge-Kutta argument for convergence order only holds strictly for the first extrapolation interval $[t_0, t_0 + h]$. From the second interval onward, the RK step solves an estimated IVP, and begins to accumulate a global estimation error not bounded by the convergence order (an effect termed "Lady Windermere's fan" by Wanner [16]). Should a probabilistic solver aim to faithfully reproduce this imperfect chain of RK solvers, or rather try to capture the accumulating global error? We investigate both options below.

**Calibration of uncertainty**    A question easily posed but hard to answer is what it means for the probability distribution returned by a probabilistic method to be well calibrated. For our Gaussian case, requiring RK order in the posterior mean determines all but one degree of freedom of an answer. The remaining parameter is the output scale of the kernel, the "error bar" of the estimate. We offer a relatively simple statistical argument below that fits this parameter based on observed values of $f$.

We can now proceed to the main results. In the following, we consider extrapolation algorithms based on Gaussian process priors with vanishing prior mean function, noise-free observation model ($\Lambda = 0$ in Eq. (3)). All covariance functions in question are integrals over the kernel $k^0(\tilde{t}, \tilde{t}') = \sigma^2 \min(\tilde{t} - \tau, \tilde{t}' - \tau)$ (parameterized by scale $\sigma^2 > 0$ and off-set $\tau \in \mathbb{R}$; valid on the domain $\tilde{t}, \tilde{t}' > \tau$), the covariance of the Wiener process [17]. Such integrated Wiener processes are Gauss-Markov processes, of increasing order, so inference in these methods can be performed by filtering, at linear cost [18]. We will use the shorthands $t = \tilde{t} - \tau$ and $t' = \tilde{t}' - \tau$ for inputs shifted by $\tau$.

### 3.2    Gauss-Markov methods matching Euler's method

**Theorem 1.** *The once-integrated Wiener process prior $p(x) = \mathcal{GP}(x; 0, k^1)$ with*

$$k^1(t, t') = \iint_\tau^{\tilde{t}, \tilde{t}'} k^0(u, v) \, du \, dv = \sigma^2 \left( \frac{\min^3(t, t')}{3} + |t - t'| \frac{\min^2(t, t')}{2} \right) \qquad (8)$$

*choosing evaluation nodes at the posterior mean gives rise to Euler's method.*

*Proof.* We show that the corresponding Butcher tableau from Table 1 holds. After "observing" the initial value, the second observation $y_1$, constructed by evaluating $f$ at the posterior mean at $t_0$, is

$$y_1 = f\left(\mu_{|x_0}(t_0), t_0\right) = f\left(\frac{k(t_0, t_0)}{k(t_0, t_0)} x_0, t_0\right) = f(x_0, t_0), \tag{9}$$

directly from the definitions. The posterior mean after incorporating $y_1$ is

$$\mu_{|x_0, y_1}(t_0 + h) = \begin{bmatrix} k(t_0 + h, t_0) & k^\partial(t_0 + h, t_0) \end{bmatrix} \begin{bmatrix} k(t_0, t_0) & k^\partial(t_0, t_0) \\ k^\partial(t_0, t_0) & \partial k^\partial(t_0, t_0) \end{bmatrix}^{-1} \begin{pmatrix} x_0 \\ y_1 \end{pmatrix} = x_0 + h y_1. \tag{10}$$

An explicit linear algebraic derivation is available in the supplements. $\square$

### 3.3 Gauss-Markov methods matching all Runge-Kutta methods of second order

Extending to second order is not as straightforward as integrating the Wiener process a second time. The theorem below shows that this only works after moving the onset $-\tau$ of the process towards infinity. Fortunately, this limit still leads to a proper posterior probability distribution.

**Theorem 2.** *Consider the twice-integrated Wiener process prior $p(x) = \mathcal{GP}(x; 0, k^2)$ with*

$$k^2(t, t') = \iint_\tau^{\tilde{t}, \tilde{t}'} k^1(u, v) du\, dv = \sigma^2 \left( \frac{\min^5(t, t')}{20} + \frac{|t - t'|}{12} \left( (t + t') \min^3(t, t') - \frac{\min^4(t, t')}{2} \right) \right). \tag{11}$$

*Choosing evaluation nodes at the posterior mean gives rise to the RK family of second order methods in the limit of $\tau \to \infty$.*

(The twice-integrated Wiener process is a proper Gauss-Markov process for all finite values of $\tau$ and $\tilde{t}, \tilde{t}' > 0$. In the limit of $\tau \to \infty$, it turns into an improper prior of infinite local variance.)

*Proof.* The proof is analogous to the previous one. We need to show all equations given by the Butcher tableau and choice of parameters hold for any choice of $\alpha$. The constraint for $y_1$ holds trivially as in Eq. (9). Because $y_2 = f(x_0 + h\alpha y_1, t_0 + h\alpha)$, we need to show $\mu_{|x_0, y_1}(t_0 + h\alpha) = x_0 + h\alpha y_1$. Therefore, let $\alpha \in (0, 1]$ arbitrary but fixed:

$$\mu_{|x_0, y_1}(t_0 + h\alpha) = \begin{bmatrix} k(t_0 + h, t_0) & k^\partial(t_0 + h, t_0) \end{bmatrix} \begin{bmatrix} k(t_0, t_0) & k^\partial(t_0, t_0) \\ \partial k(t_0, t_0) & \partial k^\partial(t_0, t_0) \end{bmatrix}^{-1} \begin{pmatrix} x_0 \\ y_1 \end{pmatrix}$$

$$= \begin{bmatrix} \frac{t_0^3(10(h\alpha)^2 + 15h\alpha t_0 + 6t_0^2)}{120} & \frac{t_0^2(6(h\alpha)^2 + 8h\alpha t_0 + 3t_0^2)}{24} \end{bmatrix} \begin{bmatrix} t_0^5/20 & t_0^4/8 \\ t_0^4/8 & t_0^3/3 \end{bmatrix}^{-1} \begin{pmatrix} x_0 \\ y_1 \end{pmatrix}$$

$$= \begin{bmatrix} 1 - \frac{10(h\alpha)^2}{3t_0^2} & h\alpha + \frac{2(h\alpha)^2}{t_0} \end{bmatrix} \begin{pmatrix} x_0 \\ y_1 \end{pmatrix}$$

$$\xrightarrow[\tau \to \infty]{} x_0 + h\alpha y_1 \tag{12}$$

As $t_0 = \tilde{t}_0 - \tau$, the mismatched terms vanish for $\tau \to \infty$. Finally, extending the vector and matrix with one more entry, a lengthy computation shows that $\lim_{\tau \to \infty} \mu_{|x_0, y_1, y_2}(t_0 + h) = x_0 + h(1 - 1/2\alpha)y_1 + h/2\alpha y_2$ also holds, analogous to Eq. (10). Omitted details can be found in the supplements. They also include the final-step posterior covariance. Its finite values mean that this posterior indeed defines a proper GP. $\square$

### 3.4 A Gauss-Markov method matching Runge-Kutta methods of third order

Moving from second to third order, additionally to the limit towards an improper prior, also requires a departure from the policy of placing extrapolation nodes at the posterior mean.

**Theorem 3.** *Consider the thrice-integrated Wiener process prior $p(x) = \mathcal{GP}(x; 0, k^3)$ with*

$$k^3(t, t') = \iint_\tau^{\tilde{t}, \tilde{t}'} k^2(u, v) du\, dv$$

$$= \sigma^2 \left( \frac{\min^7(t, t')}{252} + \frac{|t - t'| \min^4(t, t')}{720} \left( 5 \max^2(t, t') + 2tt' + 3 \min^2(t, t') \right) \right). \tag{13}$$

*Evaluating twice at the posterior mean and a third time at a specific element of the posterior covariance functions' RKHS gives rise to the entire family of RK methods of third order, in the limit of $\tau \to \infty$.*

*Proof.* The proof progresses entirely analogously as in Theorems 1 and 2, with one exception for the term where the mean does not match the RK weights exactly. This is the case for $y_3 = x_0 + h\left[(v - {v(v-u)}/{u(2-3u)})y_1 + {v(v-u)}/{u(2-3u)}y_2\right]$ (see Table 1). The weights of $Y$ which give the posterior mean at this point are given by $kK^{-1}$ (cf. Eq. (3)), which, in the limit, has value (see supp.):

$$\lim_{\tau \to \infty} \left[ k(t_0 + hv, t_0) \quad k^\partial(t_0 + hv, t_0) \quad k^\partial(t_0 + hv, t_0 + hu) \right] K^{-1}$$

$$= \left[1 \quad h\left(v - \tfrac{v^2}{2u}\right) \quad h\tfrac{v^2}{2u}\right]$$

$$= \left[1 \quad h\left(v - \tfrac{v(v-u)}{u(2-3u)} - \tfrac{\mathbf{v(3v-2)}}{\mathbf{2(3u-2)}}\right) \quad h\left(\tfrac{v(v-u)}{u(2-3u)} + \tfrac{\mathbf{v(3v-2)}}{\mathbf{2(3u-2)}}\right)\right]$$

$$= \left[1 \quad h\left(v - \tfrac{v(v-u)}{u(2-3u)}\right) \quad h\left(\tfrac{v(v-u)}{u(2-3u)}\right)\right] + \left[0 \quad -h\tfrac{v(3v-2)}{2(3u-2)} \quad h\tfrac{v(3v-2)}{2(3u-2)}\right] \quad (14)$$

This means that the final RK evaluation node does not lie at the posterior mean of the regressor. However, it can be produced by adding a correction term $w(v) = \mu(v) + \varepsilon(v)(y_2 - y_1)$ where

$$\varepsilon(v) = \frac{v}{2}\frac{3v - 2}{3u - 2} \quad (15)$$

is a second-order polynomial in $v$. Since $k$ is of third or higher order in $v$ (depending on the value of $u$), $w$ can be written as an element of the thrice integrated Wiener process' RKHS [19, §6.1]. Importantly, the final extrapolation weights $b$ under the limit of the Wiener process prior again match the RK weights exactly, regardless of how $y_3$ is constructed. $\square$

We note in passing that Eq. (15) vanishes for $v = 2/3$. For this choice, the RK observation $y_2$ is generated exactly at the posterior mean of the Gaussian process. Intriguingly, this is also the value for $\alpha$ for which the posterior variance at $t_0 + h$ is minimized.

### 3.5 Choosing the output scale

The above theorems have shown that the first three families of Runge-Kutta methods can be constructed from repeatedly integrated Wiener process priors, giving a strong argument for the use of such priors in probabilistic numerical methods. However, requiring this match to a specific Runge-Kutta family in itself does not yet uniquely identify a particular kernel to be used: The posterior mean of a Gaussian process arising from noise-free observations is independent of the output scale (in our notation: $\sigma^2$) of the covariance function (this can also be seen by inspecting Eq. (3)). Thus, the parameter $\sigma^2$ can be chosen independent of the other parts of the algorithm, without breaking the match to Runge-Kutta. Several algorithms using the observed values of $f$ to choose $\sigma^2$ without major cost overhead have been proposed in the regression community before [e.g. 20, 21]. For this particular model an even more basic rule is possible: A simple derivation shows that, in all three families of methods defined above, the posterior belief over $\partial^s x/\partial t^s$ is a Wiener process, and the posterior mean function over the $s$-th derivative after all $s$ steps is a constant function. The Gaussian model implies that the expected distance of this function from the (zero) prior mean should be the marginal standard deviation $\sqrt{\sigma^2}$. We choose $\sigma^2$ such that this property is met, by setting $\sigma^2 = \left[\partial^s \mu_s(t)/\partial t^s\right]^2$.

Figure 1 shows conceptual sketches highlighting the structure of GMRK methods. Interestingly, in both the second- and third-order families, our proposed priors are improper, so the solver can not actually return a probability distribution until after the observation of all $s$ gradients in the RK step.

**Some observations** We close the main results by highlighting some non-obvious aspects. First, it is intriguing that higher convergence order results from repeated integration of Wiener processes. This repeated integration simultaneously adds to and weakens certain prior assumptions in the implicit (improper) Wiener prior: $s$-times integrated Wiener processes have marginal variance $k^s(t,t) \propto t^{2s+1}$. Since many ODEs (e.g. linear ones) have solution paths of values $\mathcal{O}(\exp(t))$, it is tempting to wonder whether there exists a limit process of "infinitely-often integrated" Wiener processes giving natural coverage to this domain (the results on a linear ODE in Figure 1 show how the polynomial posteriors cannot cover the exponentially diverging true solution). In this context,

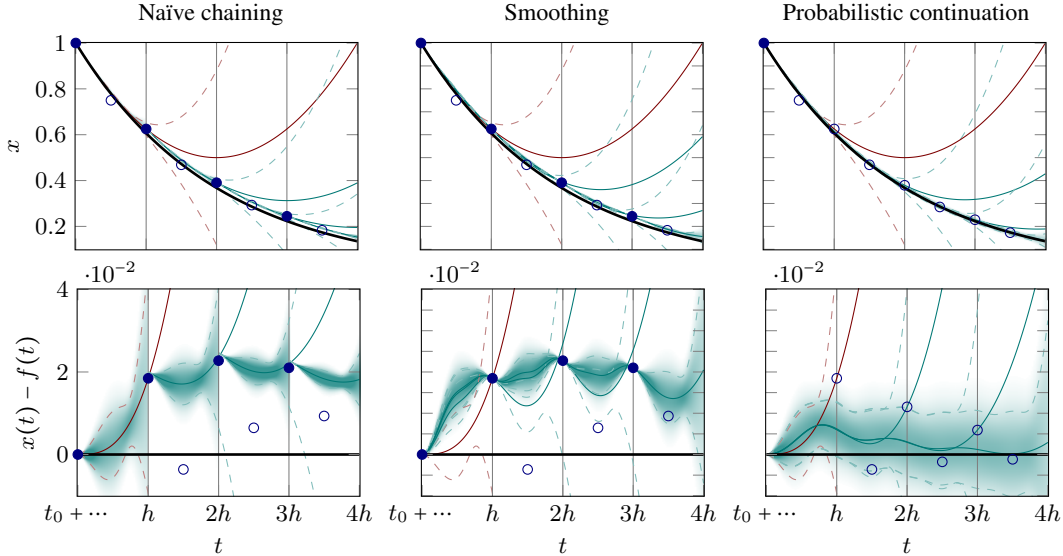

Figure 2: Options for the continuation of GMRK methods after the first extrapolation step (red). All plots use the midpoint method and $h = 1$. Posterior after two steps (same for all three options) in red (mean, $\pm 2$ standard deviations). Extrapolation after 2, 3, 4 steps (gray vertical lines) in green. Final probabilistic prediction as green shading. True solution to (linear) ODE in black. Observations of $x$ and $\dot{x}$ marked by solid and empty blue circles, respectively. Bottom row shows the same data, plotted relative to true solution, at higher y-resolution.

it is also noteworthy that $s$-times integrated Wiener priors incorporate the lower-order results for $s' < s$, so "highly-integrated" Wiener kernels can be used to match finite-order Runge-Kutta methods. Simultaneously, though, sample paths from an $s$-times integrated Wiener process are almost surely $s$-times differentiable. So it seems likely that achieving good performance with a Gauss-Markov-Runge-Kutta solver requires trading off the good marginal variance coverage of high-order Markov models (i.e. repeatedly integrated Wiener processes) against modelling non-smooth solution paths with lower degrees of integration. We leave this very interesting question for future work.

## 4   Experiments

Since Runge-Kutta methods have been extensively studied for over a century [11], it is not necessary to evaluate their estimation performance again. Instead, we focus on an open conceptual question for the further development of probabilistic Runge-Kutta methods: If we accept high convergence order as a prerequisite to choose a probabilistic model, how should probabilistic ODE solvers continue *after* the first $s$ steps? Purely from an inference perspective, it seems unnatural to introduce new evaluations of $x$ (as opposed to $\dot{x}$) at $t_0 + nh$ for $n = 1, 2, \ldots$. Also, with the exception of the Euler case, the posterior covariance after $s$ evaluations is of such a form that its renewed use in the next interval will not give Runge-Kutta estimates. Three options suggest themselves:

**Naïve Chaining**   One could simply re-start the algorithm several times as if the previous step had created a novel IVP. This amounts to the classic RK setup. However, it does not produce a joint "global" posterior probability distribution (Figure 2, left column).

**Smoothing**   An ad-hoc remedy is to run the algorithm in the "Naïve chaining" mode above, producing $N \times s$ gradient observations and $N$ function evaluations, but then compute a joint posterior distribution by using the first $s$ gradient observations and 1 function evaluation as described in Section 3, then using the remaining $s(N - 1)$ gradients and $(N - 1)$ function values as in standard GP inference. The appeal of this approach is that it produces a GP posterior whose mean goes through the RK points (Figure 2, center column). But from a probabilistic standpoint it seems contrived. In particular, it produces a very confident posterior covariance, which does not capture global error.

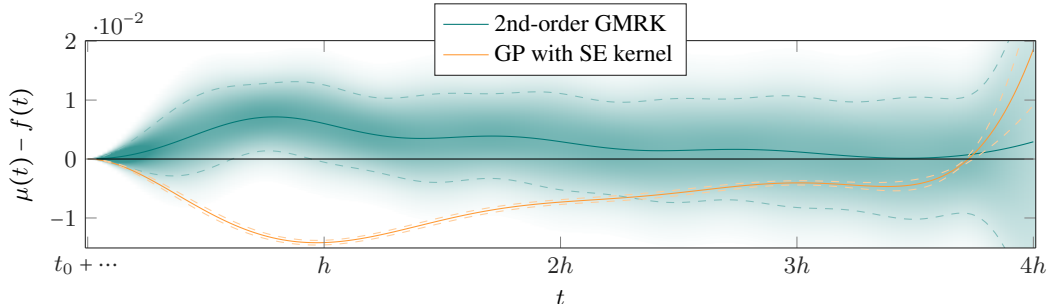

Figure 3: Comparison of a 2nd order GMRK method and the method from [6]. Shown is error and posterior uncertainty of GMRK (green) and SE kernel (orange). Dashed lines are +2 standard deviations. The SE method shown used the best out of several evaluated parameter choices.

**Continuing after $s$ evaluations**  Perhaps most natural from the probabilistic viewpoint is to break with the RK framework after the first RK step, and simply continue to collect gradient observations—either at RK locations, or anywhere else. The strength of this choice is that it produces a continuously growing marginal variance (Figure 2, right). One may perceive the departure from the established RK paradigm as problematic. However, we note again that the core theoretical argument for RK methods is only strictly valid in the first step, the argument for iterative continuation is a lot weaker.

Figure 2 shows exemplary results for these three approaches on the (stiff) linear IVP $\dot{x}(t) = -\frac{1}{2}x(t)$, $x(0) = 1$. Naïve chaining does not lead to a globally consistent probability distribution. Smoothing does give this global distribution, but the "observations" of function values create unnatural nodes of certainty in the posterior. The probabilistically most appealing mode of continuing inference directly offers a naturally increasing estimate of global error. At least for this simple test case, it also happens to work better in practice (note good match to ground truth in the plots). We have found similar results for other test cases, notably also for non-stiff linear differential equations. But of course, probabilistic continuation breaks with at least the traditional mode of operation for Runge-Kutta methods, so a closer theoretical evaluation is necessary, which we are planning for a follow-up publication.

**Comparison to Square-Exponential kernel**  Since all theoretical guarantees are given in forms of upper bounds for the RK methods, the application of different GP models might still be favorable in practice. We compared the continuation method from Fig. 2 (right column) to the ad-hoc choice of a square-exponential (SE) kernel model, which was used by Hennig and Hauberg [6] (Fig. 3). For this test case, the GMRK method surpasses the SE-kernel algorithm both in accuracy and calibration: its mean is closer to the true solution than the SE method, and its error bar covers the true solution, while the SE method is over-confident. This advantage in calibration is likely due to the more natural choice of the output scale $\sigma^2$ in the GMRK framework.

## 5 Conclusions

We derived an interpretation of Runge-Kutta methods in terms of the limit of Gaussian process regression with integrated Wiener covariance functions, and a structured but nontrivial extrapolation model. The result is a class of probabilistic numerical methods returning Gaussian process posterior distributions whose means can match Runge-Kutta estimates exactly.

This class of methods has practical value, particularly to machine learning, where previous work has shown that the probability distribution returned by GP ODE solvers adds important functionality over those of point estimators. But these results also raise pressing open questions about probabilistic ODE solvers. This includes the question of how the GP interpretation of RK methods can be extended beyond the 3rd order, and how ODE solvers should proceed after the first stage of evaluations.

**Acknowledgments**

The authors are grateful to Simo Särkkä for a helpful discussion.

## Footnotes

[1]In this work, we only address so-called *explicit* RK methods (shortened to "Runge-Kutta methods" for simplicity). These are the base case of the extensive theory of RK methods. Many generalizations can be found in [11]. Extending the probabilistic framework discussed here to the wider Runge-Kutta class is not trivial.

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
