[Supplementary Material]

# Probabilistic ODE Solvers with Runge-Kutta Means
# — Supplementary Material —

**Michael Schober**
MPI for Intelligent Systems
Tübingen, Germany
mschober@tue.mpg.de

**David Duvenaud**
Department of Engineering
Cambridge University
dkd23@cam.ac.uk

**Philipp Hennig**
MPI for Intelligent Systems
Tübingen, Germany
phennig@tue.mpg.de

This document contains derivation steps omitted in the main paper. Additionally, the website[1] to this publication contains MATLAB Symbolic Math Toolbox code which was used to obtain the lengthy derivations.

## A    Multivariate extension

The GMRK model can be extended to the multivariate case analogously to Runge-Kutta methods. All equations in the RK framework also work with vector-valued function values, and all derivations presented in the paper and in this supplement carry over to the non-scalar case without modification: Consider dimension $j \in \{1, \dots, N\}$. The projected outputs are the same as if $j$ were an independent one-dimensional problem, which can be modeled with a separate Gaussian process. For a joint notation, vectorize the $N$ dimensions with a Kronecker product: if $k(t, t')$ is an one-dimensional covariance function, the function

$$\bar{k}(\boldsymbol{t}, \boldsymbol{t}') = \boldsymbol{D}_{ij} k(t_i, t_j'), \tag{1}$$

where $\boldsymbol{D}$ is an $N \times N$ positive semi-definite matrix, defines a covariance function over $N$ dimensions, if $\boldsymbol{t}$ and $\boldsymbol{t}'$ are $N$ dimensional. Choosing $\boldsymbol{D} = \boldsymbol{I}$ results in a $N$-dimensional GP model where output dimensions are independent of each other as required.

## B    Covariance functions of integrated Wiener processes

It was observed that integrated Wiener processes generate RK methods of various order with higher number of integrations leading to RK methods of higher order.

Here we present the derivation of the covariance functions of the applied Wiener process kernels as well as their derivatives as needed.

### B.1    The once integrated Wiener process

The *Wiener process* covariance function is given by

$$k_{WP}(t, t') = \sigma^2 \min(t, t') \tag{2}$$

It is only defined for $t, t' > 0$. Integration with respect to both arguments leads to the *once integrated Wiener process* which is once differentiable. Its covariance function is

$$k_1(t, t') = \int_0^t du \int_0^{t'} dv \ \sigma^2 \min(u, v)$$

$$= \sigma^2 \int_0^t du \int_0^{t'} dv \ \min(u, v)$$

$$\stackrel{t \geq t'}{=} \sigma^2 \left( \int_{t'}^t du \int_0^{t'} dv \ v + 2 \int_0^{t'} du \int_0^u dv \ v \right)$$

$$= \sigma^2 \left( \int_{t'}^t du \ \frac{1}{2} t'^2 + 2 \int_0^{t'} du \ \frac{1}{2} u^2 \right)$$

$$= \sigma^2 \left( \frac{1}{2} (t - t') t'^2 + \frac{1}{3} (t'^3) \right)$$

$$= \sigma^2 \left( \frac{\min^3(t, t')}{3} + |t - t'| \frac{\min^2(t, t')}{2} \right) \tag{3}$$

where $t, t'$ were replaced with $\min(t, t')$ and $\max(t, t')$ at the last step.

The necessary derivatives of this kernel are

$$k^\partial(t, t') = \sigma^2 \begin{cases} t < t' : & \frac{t^2}{2} \\ t > t' : & \left( tt' - \frac{t'^2}{2} \right) \end{cases} \tag{4}$$

$$\partial k^\partial(t, t') = \sigma^2 \min(t, t') \qquad\qquad = k_{WP}(t, t'). \tag{5}$$

## B.2 The twice integrated Wiener process

Iterating this process leads to the *twice integrated Wiener process*. Its covariance function is

$$k_2(t, t') = \sigma^2 \left( \int_0^t du \int_0^{t'} dv \ \frac{\min^3(u, v)}{3} + |u - v| \frac{\min^2(u, v)}{2} \right)$$

$$\stackrel{t \geq t'}{=} \sigma^2 \left( \left( \int_{t'}^t du \int_0^{t'} dv \ (u - v) \frac{v^2}{2} + \frac{v^3}{3} \right) + \left( \int_0^{t'} du \int_0^u dv \ (u - v) \frac{v^2}{2} + \frac{v^3}{3} \right) \right.$$

$$\left. + \left( \int_0^v du \int_0^{t'} dv \ (v - u) \frac{u^2}{2} + \frac{u^3}{3} \right) \right)$$

$$\vdots$$

$$= \sigma^2 \left( \frac{\min^5(t, t')}{20} + \frac{|t - t'|}{12} \left( (t + t') \min^3(t, t') - \frac{\min^4(t, t')}{2} \right) \right) \tag{6}$$

Derivatives of this kernel are

$$k^\partial(t, t') = \sigma^2 \begin{cases} t > t' : & \left( \frac{t'^2}{24} (t'^2 - 4tt' + 6t^2) \right) \\ t \leq t' : & \left( -\frac{t^4}{24} + \frac{t' t^3}{6} \right) \end{cases} \tag{7}$$

$$\partial k^\partial(t, t') = \sigma^2 \left( \frac{\min^3(t, t')}{3} + |t - t'| \frac{\min^2(t, t')}{2} \right) \qquad = k_1(t, t'). \tag{8}$$

## B.3 The thrice integrated Wiener process

Similarly, the *thrice integrated Wiener process* is obtained by

$$k_3(t, t') = \sigma^2 \left( \int_0^t du \int_0^{t'} dv \ \frac{\min^5(u, v)}{20} + \frac{|u - v|}{12} \left( (u + v) \min^3(u, v) - \frac{\min^4(u, v)}{2} \right) \right)$$

$$\vdots$$

$$= \sigma^2 \left( \frac{\min^7(t, t')}{252} + \frac{|t - t'| \min^4(t, t')}{720} \left( 5 \max^2(t, t') + 2tt' + 3 \min^2(t, t') \right) \right) \tag{9}$$

Omitted steps are similar as in the derivation of (3) and (6).

Its derivatives are given by

$$k^\partial(t,t') = \sigma^2 \begin{cases} t > t': & \left( \frac{t'^3}{720} \left( 20t^3 - 15t^2 t' + 6tt'^2 - t'^3 \right) \right) \\ t \le t': & \left( \frac{t^4}{720} \left( 15t'^2 - 6tt' + t^2 \right) \right) \end{cases} \tag{10}$$

$$\partial k^\partial(t,t') = \sigma^2 \left( \frac{\min^5(t,t')}{20} + \frac{|t-t'|}{12} \left( (t+t') \min^3(t,t') - \frac{\min^4(t,t')}{2} \right) \right) = k_2(t,t'). \tag{11}$$

## C   Posterior predictive GP distributions

In order to build GMRK methods it is necessary to compute closed forms of the resulting posterior mean and covariance functions after $s$ evaluations. Forms are given below. In cases where a derivation is omitted, results were obtained with MATLAB's Symbolic Math Toolbox. Code is available online.

### C.1   Posterior predictive mean and covariance functions of the once integrated WP

Below are the formulas of the posterior mean and covariance of the once integrated WP after each step.

$$\mu_{|x_0}(t) = \frac{k(t,t_0)}{k(t_0,t_0)} x_0$$

$$= \frac{t_0^3/3 + |t-t_0| t_0^2/2}{t_0^3/3} x_0$$

$$= \left( 1 + |t-t_0| \frac{3}{2t_0} \right) x_0 \tag{12}$$

$$\mu_{|x_0,y_1}(t) = \begin{bmatrix} k(t,t_0) & k^\partial(t,t_0) \end{bmatrix} \underbrace{\begin{bmatrix} k(t_0,t_0) & k^\partial(t_0,t_0) \\ \partial k(t_0,t_0) & \partial k^\partial(t_0,t_0) \end{bmatrix}}_{=:K}^{-1} \begin{pmatrix} x_0 \\ y_1 \end{pmatrix}$$

$$= \begin{cases} t \ge t_0: & \frac{1}{|K|} \begin{bmatrix} t^2 t_0/2 - t^3/6 & t^2/2 \end{bmatrix} \begin{bmatrix} t_0 & -t_0^2/2 \\ -t_0^2/2 & t_0^3/3 \end{bmatrix} \begin{pmatrix} x_0 \\ y_1 \end{pmatrix} \\ t < t_0: & \frac{1}{|K|} \begin{bmatrix} t t_0^2/2 - t_0^3/6 & tt_0 - t_0^2/2 \end{bmatrix} \begin{bmatrix} t_0 & -t_0^2/2 \\ -t_0^2/2 & t_0^3/3 \end{bmatrix} \begin{pmatrix} x_0 \\ y_1 \end{pmatrix} \end{cases}$$

$$\vdots$$

$$= \begin{cases} t \ge t_0: & x_0 + (t-t_0)y_1 \\ t < t_0: & \frac{3t^2 t_0 - 2t^3}{t_0^3} x_0 - \frac{t^2 t_0 - t^3}{t_0^2} y_1 \end{cases} \tag{13}$$

Eqs. (12) and (13) also complete the proof of Theorem 1 by observing that $y_1 = f(x_0,t_0) = f(\mu_{|x_0}(t_0),t_0)$ and $x_1 = x_0 + hy_1 = \mu_{|x_0,y_1}(t_0 + h)$, which match Euler's method.

Without loss of generality, we can assume that $t' \leq t$. With this convention the posterior covariances functions are

$$
\begin{aligned}
k_{|x_0}(t,t') &= k^1(t,t') - \frac{k^1(t,t_0)k^1(t_0,t')}{k^1(t_0,t_0)} \\
&= \left( \frac{\min^3(t,t')}{3} + |t-t'| \frac{\min^2(t,t')}{2} \right) \\
&\quad - \frac{1}{24t_0^3} \left( \min^2(t,t_0)\min^2(t_0,t')(t+t_0+2|t-t_0|)(t'+t_0+2|t'-t_0|) \right)
\end{aligned}
\tag{14}
$$

$$
k_{|x_0,y_1}(t,t') = k^1(t,t') - \begin{bmatrix} k(t,t_0) & k^\partial(t,t_0) \end{bmatrix} K^{-1} \begin{bmatrix} k(t_0,t') \\ \partial k(t_0,t') \end{bmatrix}
$$

$$
\vdots
$$

$$
= \begin{cases} t,t' > t_0 : & \frac{(t_0-t')^2(3t-t'-2t_0)}{6} \\ t > t_0 \geq t' : & 0 \\ t,t' \leq t_0 : & \frac{t'^2(t_0-t)^2(3tt_0-t't_0-2tt')}{6t_0^3} \end{cases}
\tag{15}
$$

## C.2 Predictive mean and covariance of the twice integrated WP

Below are the formulas for the posterior mean for the twice integrated WP and the generic 2-stage RK method. Throughout, we write $\mu(t) = \mu(t_0+s)$ with appropriate $s \in \mathbb{R}$, which will simplify formulas significantly. Furthermore, we omit stating the generating formulas and intermediate steps as the former are analogous to the ones in Sec. C.1 and the latter were performed with MATLAB's Symbolic Math Toolbox.

$$
\mu_{|x_0}(t_0+s) = \left[ 1 + \frac{5s}{2t_0} + \frac{5s^2}{3t_0^2} + \mathbb{I}_{(-\infty,s)}\left( \frac{s^5}{6t_0^5} \right) \right] x_0
\tag{16}
$$

$$
\begin{aligned}
\mu_{|x_0,y_1}(t_0+s) = & \left[ 1 - \frac{10s^2}{3t_0^2} + \mathbb{I}_{(-\infty,s)}\left( \frac{5s^4}{t_0^4} + \frac{8s^5}{3t_0^5} \right) \right] x_0 \\
& + \left[ s + \frac{2s^2}{t_0} - \mathbb{I}_{(-\infty,s)}\left( \frac{2s^4}{t_0^3} + \frac{s^5}{t_0^4} \right) \right] y_0
\end{aligned}
\tag{17}
$$

$$
\lim_{t_0 \to \infty} \mu_{|x_0,y_1,y_2}(t_0+s) = x_0 + \left( s - \frac{s^2}{2h\alpha} \right) y_1 + \frac{s^2}{2h\alpha} y_2
\tag{18}
$$

As was the case for the posterior mean, we will write the posterior covariance function as $k(t,t') = k(t_0+s, t_0+s')$ while also assuming w.l.o.g. that $s' \leq s$. The posterior covariance functions are then given by:

$$
\begin{aligned}
k_{|x_0}(t_0+s, t_0+s') = & \frac{ss'}{48} t_0^3 + \left( ss'^2 + s^2s' \right) \frac{t_0^2}{24} + \frac{s^2 s'^2}{9} t_0 \\
& + \left( \frac{|s|^5}{240} + \frac{s^2 s'^3}{24} + \frac{s^3 s'^2}{24} - \frac{ss'^4}{48} - \frac{s^4 s'}{48} + \frac{|s'|^5}{240} - \frac{|s'-s|^5}{240} \right) t_0^0 \\
& - \left( ss'^5 + s^5 s' - s|s'|^5 - s'|s|^5 \right) \frac{t_0^{-1}}{96} - \left( s^2 s'^5 + s^5 s'^2 - s^2|s'|^5 - s'^2|s|^5 \right) \frac{t_0^{-2}}{144} \\
& - \left( s^5 - |s|^5 \right)\left( s'^5 - |s'|^5 \right) \frac{t_0^{-5}}{2880} \\
& \xrightarrow[t_0 \to \infty]{} \infty
\end{aligned}
\tag{19}
$$

$$k_{|x_0,y_1}(t_0 + s, t_0 + s') = \begin{cases} s,s' > 0: & \frac{s'^5}{240} + \frac{s^2 s'^3}{24} + \frac{s^3 s'^2}{24} - \frac{|s'-s|^5}{240} + \frac{s^5}{240} - \frac{ss'^4}{48} - \frac{s^4 s'}{48} + \frac{s^2 s'^2}{36}t_0 \\[2mm] s > 0 \geq s': & \frac{s^2 s'^2 (s'+t_0)^3}{36 t_0^2} \\[2mm] s,s' \leq 0: & \frac{s^2 s'^3}{24} - \frac{s'^5}{240} + \frac{s^3 s'^2}{24} - \frac{|s'-s|^5}{240} - \frac{s^5}{240} + \frac{ss'^4}{48} + \frac{s^4 s'}{48} \\[1mm] & + \frac{s^2 s'^2}{36} t_0 + \frac{s^2 s'^2(s^2+s'^2)}{12 t_0} + \frac{s^2 s'^2 (s^3 + s'^3)}{36 t_0^2} - \frac{s^4 s'^4 (s'+s)}{24 t_0^4} \\[1mm] & - \frac{s^4 s'^4}{12 t_0^3} - \frac{s^5 s'^5}{45 t_0^5} \end{cases}$$

$$\xrightarrow[t_0 \to \infty]{} \infty$$

(20)

For the final posterior covariance, it is also necessary to distinguish between the cases whether $s, s' \geq h\alpha$, $s \geq h\alpha > s'$ and $s, s' \leq h\alpha$.

$$k_{|x_0,y_1,y_2}(t_0 + s, t_0 + s') = \begin{cases} s,s' > (h\alpha): & \big[\big(8s'^5 - 40ss'^4 + 80(s^2 s'^3 + (h\alpha)^2(s^2 s' + ss'^2)) \\ & - 20(h\alpha)^3(s^2 + s'^2) - 160(h\alpha)s^2 s'^2\big)t_0 \\ & - 15(h\alpha)^6 + 60(h\alpha)^5(s + s') - 90(h\alpha)^4(s^2 + s'^2) \\ & + 24(h\alpha)s'^5 + 360(h\alpha)^3(ss'^2 + s^2 s') \\ & - 120(h\alpha)ss'^4 - 540(h\alpha)^2 s^2 s'^2 + 240(h\alpha)s^2 s'^3 \\ & - 240(h\alpha)^4 ss'\big]\big[960t_0 + 2880h\alpha\big]^{-1} \\[2mm] s > (h\alpha) \geq s' > 0: & \big[\big(20(s^2 s'^4 - (h\alpha)^4 s'^2) + 80(h\alpha)^3 ss'^2 + 8(h\alpha)s'^5 \\ & - 40((h\alpha)^2 s^2 s'^2 + (h\alpha)ss'^4)\big)t_0 + 24(h\alpha)^2 s'^5 \\ & + 15(h\alpha)^3 s'^4 - 60(h\alpha)^4 s'^3 - 180(h\alpha)^2 ss'^4 \\ & + 240(h\alpha)^3 ss'^3 - 120(h\alpha)^2 s^2 s'^3 + 90(h\alpha)s^2 s'^4\big] \\ & \big[960t_0 h\alpha + 2880(h\alpha)^2\big]^{-1} \\[2mm] s > (h\alpha) > 0 \geq s': & \big[(h\alpha)s'^2(s'+t_0)^3\big(4(h\alpha)s - 2s^2 - (h\alpha)^2\big)\big] \\ & \big[48t_0^2(t_0 + 3(h\alpha))\big]^{-1} \\[2mm] (h\alpha) \geq s, s' > 0: & \big[\big(80(h\alpha)^2 s^2 s'^2((h\alpha) - s) - 40(h\alpha)^2 ss'^4 \\ & + 8(h\alpha)^2 s'^5 + 20(h\alpha)s^2 s'^2(s^2 + s'^2)\big)t_0 \\ & - 15s^4 s'^4 + 24(h\alpha)^3 s'^5 - 120(h\alpha)^3 ss'^4 \\ & + 240(h\alpha)^2 s^2 s'^3((h\alpha) - s) \\ & + 60(h\alpha)s^3 s'^3(s + s')\big] \\ & \big[960t_0(h\alpha)^2 + 2880(h\alpha)^3\big]^{-1} \\[2mm] (h\alpha) \geq s > 0 \geq s': & \big[s^2 s'^2(s'+t_0)^3(s - 2(h\alpha))^2\big] \\ & \big[48t_0^2(h\alpha)(t_0 + 3(h\alpha))\big]^{-1} \\[2mm] s,s' \leq 0: & -\big[s^2(s'+t_0)^3\big(8s^3 s'^2 - 9s^3 s' t_0 + 3s^3 t_0^2 \\ & + 15s^2 s' t_0(s' - t_0) - 10s'^2 t_0^3\big)\big]\big[360t_0^5\big]^{-1} \\ & - \big[s^2 s'^2(s+t_0)^3(s'+t_0)^3\big]\big[36t_0^4(t_0 + 3(h\alpha))\big]^{-1} \end{cases}$$

(21)

$$\lim_{t_0 \to \infty} k_{|x_0,y_1,y_2}(t_0 + s, t_0 + s') = \begin{cases} s,s' > (h\alpha): & \big[s'^3/12 - (h\alpha)s'^2/6 + (h\alpha)^2 s'/12 - (h\alpha)^3/48\big]s^2 \\ & \big[(h\alpha)^2 s'^2/12 - s'^4/24\big]s + s'^5/120 - (h\alpha)^3 s'^2/48 \\[2mm] s > (h\alpha) \geq s' > 0: & \big[s'^2\big(20(h\alpha)^3 s - 10(h\alpha)s((h\alpha)s + s'^2) \\ & + 2(h\alpha)s'^3 + 5(s^2 s'^2 - (h\alpha)^4)\big)\big] \\ & \big[240(h\alpha)\big]^{-1} \\[2mm] s > (h\alpha) > 0 \geq s': & -1/48\big[(h\alpha)s'^2((h\alpha)^2 - 4(h\alpha)s + 2s^2)\big] \\[1mm] (h\alpha) \geq s, s' > 0: & \big[s'^2\big(20(h\alpha)s^2((h\alpha) - s) - 10(h\alpha)ss'^2 \\ & + 2(h\alpha)s'^3 + 5s^2(s^2 + s'^2)\big)\big]\big[240(h\alpha)\big]^{-1} \\[2mm] (h\alpha) \geq s > 0 \geq s': & \big[s^2 s'^2(s - 2(h\alpha))^2\big]\big[48(h\alpha)\big]^{-1} \\[1mm] s,s' \leq 0: & -1/120\big[s^2(s^3 - 5s^2 s' + 10ss'2 - 10(h\alpha)s'^2)\big] \end{cases}$$

(22)

Eq. (22) concludes the proof of Theorem 2 as it shows that the covariance function after the RK step is finite for all values of $s, s'$.

## C.3 Posterior predictive mean and covariance of the thrice integrated WP

Here we list the equations of posterior mean and covariance for the thrice integrated WP and the generic 3-stage RK method. The same structure as in Sec. C.2 was applied.

$$\lim_{t_0 \to \infty} \mu_{|x_0}(t_0 + s) = x_0 \tag{23}$$

$$\lim_{t_0 \to \infty} \mu_{|x_0,y_1}(t_0 + s) = x_0 + sy_1 \tag{24}$$

$$\lim_{t_0 \to \infty} \mu_{|x_0,y_1,y_2}(t_0 + s) = x_0 + \left(s - \frac{s^2}{2hu}\right) y_1 + \frac{s^2}{2hu} y_2 \tag{25}$$

$$\lim_{t_0 \to \infty} \mu_{|x_0,y_1,y_2,y_3}(t_0 + s) = x_0 + \left(s - \frac{h(\frac{s^2 u}{2} + \frac{s^2 v}{2}) - \frac{s^3}{3}}{h^2 uv}\right) y_1 \tag{26}$$

$$+ \left(\frac{s^2(2s - 3hv)}{6h^2 u(u - v)}\right) y_2 + \left(-\frac{s^2(2s - 3hu)}{6h^2 v(u - v)}\right) y_3 \tag{27}$$

As in the case of the twice integrated Wiener process, the covariance function is infinite for $\lim_{t_0 \to \infty}$. Therefore, we only list the final posterior covariance function.

For $s, s' > hv > hu > 0$:

$$\lim_{t_0 \to \infty} k_{|x_0,y_1,y_2,y_3}(t_0 + s, t_0 + s') =$$
$$\left\{\left[-21h^5 u(s^2 + s'^2) + 14h^4(s^3 + s'^3)\right] v^5 + \left[-21h^5 u^2(s^2 + s'^2) + 14h^4 u(s^3 + s'^3)\right.\right.$$
$$\left. + 126h^4 u(s^2 s' + ss'^2) - 84h^3(s^3 s' + ss'^3)\right] v^4 + \left[-21h^5 u^3(s^2 + s'^2) + 14h^4 u^2(s^3 + s'^3)\right.$$
$$\left. + 126h^4 u^2(s^2 s' + ss'^2) - 84h^3 u(s^3 s' + ss'^3) - 630h^3 u s^2 s'^2 + 210h^2(s^3 s'^2 + s^2 s'^3)\right] v^3$$
$$+ \left[-21h^5 u^4(s^2 + s'^2) + 14h^4 u^3(s^3 + s'^3) + 126h^4 u^3(s^2 s' + ss'^2) - 84h^3 u^2(s^3 s' + ss'^3)\right.$$
$$\left. - 252h^3 u^2 s^2 s'^2 + 378h^2 u(s^3 s'^2 + s^2 s'^3) - 392h s^3 s'^3\right] v^2 + \left[14h^4 u^4(s^3 + s'^3)\right.$$
$$\left. - 84h^3 u^3(s^3 s' + ss'^3) + 126h^2 u^2(s^3 s'^2 - s^2 s'^2 + s^2 s'^3) - 224hu s^3 s'^3 + 210 s^3 s'^4 - 126 s^2 s'^5\right.$$
$$\left. + 42 ss'^6 - 6s'^7\right] v + 42h^2 u^3(s^3 s'^2 + s^2 s'^3) - 56hu^2 s^3 s'^3 \right\}/(30240v) \tag{28}$$

For $s > hv \geq s' > hu > 0$:

$$\lim_{t_0 \to \infty} k_{|x_0,y_1,y_2,y_3}(t_0 + s, t_0 + s') =$$
$$\left\{\left(21h^7 u s'^2 - 14h^6 s'^3\right) v^6 + \left(-126 suh^6 s'^2 + 84 sh^5 s'^3\right) v^5\right.$$
$$+ (315uh^5 s^2 s'^2 - 210h^4 s^2 s'^3) v^4$$
$$+ (-378h^5 s^2 s'^2 u^2 - 168h^4 s^3 s'^2 u + 252h^4 s^2 s'^3 u + 112h^3 s^3 s'^3) v^3$$
$$+ (-21h^7 s^2 u^5 - 21h^7 s'^2 u^5 + 126h^6 s^2 s' u^4 + 126h^6 ss'^2 u^4 - 126h^5 s^2 s'^2 u^3 + 252h^4 s^3 s'^2 u^2$$
$$+ 252h^4 s^2 s'^3 u^2 - 168h^3 s^3 s'^3 u - 315h^3 s^2 s'^4 u + 126h^2 s^2 s'^5 - 42h^2 ss'^6 + 6h^2 s'^7) v^2$$
$$+ (14h^6 s^3 u^5 - 84h^5 s^3 s' u^4 - 126h^5 s^2 s'^2 u^4 - 84h^5 ss'^3 u^4 + 84h^4 s^3 s'^2 u^3$$
$$+ 84h^4 s^2 s'^3 u^3 - 168h^3 s^3 s'^3 u^2 + 210h^2 s^3 s'^4 u + 42h^2 ss'^6 u - 6h^2 s'^7 u - 84h s^3 s'^5) v$$
$$+ 42h^4 s^3 s'^2 u^4 + 42h^4 s^2 s'^3 u^4 - 56h^3 s^3 s'^3 u^3 - 21h s^2 s'^6 u + 14 s^3 s'^6\right\}/(-30240h^2 v^2 + 30240 uh^2 v) \tag{29}$$

For $s > hv > hu \geq s' > 0$:

$$\lim_{t_0 \to \infty} k_{|x_0,y_1,y_2,y_3}(t_0 + s, t_0 + s') =$$

$$\frac{s'^2}{30240h^2uv}\Big[-21h^7(u^5v^2 + u^4v^3 + u^3v^4 + u^2v^5) - 21hs^2s'^4(u+v) - 126h^4s^2u^3v(hu - s')$$

$$+ 126h^6s(u^4v^2 + u^3v^3 + u^2v^4) + 14h^6s'(u^5v + u^4v^2 + u^3v^3 + u^2v^4 + uv^5) + 14s^3s'^4$$

$$+ 63h^5u^3v^2s^2 - 315h^5u^2v^3s^2 - 84h^5ss'(u^4v + u^3v^2 + u^2v^3 + uv^4) - 84h^4u^3vs^3$$

$$+ 42h^4u^4s^2(s+s') - 42h^4u^2v^2s^2s' + 42h^2uvss'^4 + 168h^4u^2v^2s^3 + 210h^4uv^3s^2s'$$

$$- 56h^3u^2s^3s'(u-v) - 112h^3uv^2s^3s' - 6h^2uvs'^5\Big] \quad (30)$$

For $s > hv > hu > 0 \geq s'$:

$$\lim_{t_0 \to \infty} k_{|x_0,y_1,y_2,y_3}(t_0 + s, t_0 + s') =$$

$$\frac{hs'^2}{4320v}\Big[(-3h^4u^4v^2 - 3h^4u^3v^3 - 3h^4u^2v^4 - 3h^4uv^5 + 18h^3su^3v^2 + 18h^3su^2v^3 + 18h^3suv^4$$

$$+ 2s'h^3u^4v + 2s'h^3u^3v^2 + 2s'h^3u^2v^3 + 2s'h^3uv^4 + 2s'h^3v^5 - 18h^2s^2u^3v + 9h^2s^2u^2v^2$$

$$- 45h^2s^2uv^3 - 12s'h^2su^3v - 12s'h^2su^2v^2 - 12s'h^2suv^3 - 12s'h^2sv^4 + 6hs^3u^3$$

$$- 12hs^3u^2v + 24hs^3uv^2 + 6s'hs^2u^3 + 18s'hs^2u^2v - 6s'hs^2uv^2 + 30s'hs^2v^3 - 8s's^3u^2$$

$$+ 8s's^3uv - 16s's^3v^2)\Big] \quad (31)$$

For $hv \geq s, s' > hu > 0$:

$$\lim_{t_0 \to \infty} k_{|x_0,y_1,y_2,y_3}(t_0 + s, t_0 + s') =$$

$$\Big\{(378h^5s^2s'^2u^2 - 252h^4s^3s'^2u - 252h^4s^2s'^3u + 168h^3s^3s'^3)v^3$$

$$+ (21h^7s^2u^5 + 21h^7s'^2u^5 - 126h^6s^2s'u^4 - 126h^6ss'^2u^4 + 126h^5s^2s'^2u^3$$

$$- 252h^4s^3s'^2u^2 - 252h^4s^2s'^3u^2 + 315h^3s^4s'^2u + 168h^3s^3s'^3u + 315h^3s^2s'^4u'$$

$$- 210h^2s^4s'^3 - 126h^2s^2s'^5 + 42h^2ss'^6 - 6h^2s'^7)v^2$$

$$+ (-14h^6s^3u^5 - 14h^6s'^3u^5 + 84h^5s^3s'u^4 + 126h^5s^2s'^2u^4 + 84h^5ss'^3u^4$$

$$- 84h^4s^3s'^2u^3 - 84h^4s^2s'^3u^3 + 168h^3s^3s'^3u^2 - 126h^2s^5s'^2u$$

$$- 126h^2s^5s'^2u - 210h^2s^3s'^4u - 42h^2ss'^6u + 6h^2s'^7u + 84hs^5s'^3 + 84hs^3s'^6)v$$

$$- 42h^4s^3s'^2u^4 - 42h^4s^2s'^3u^4 + 56h^3s^3s'^3u^3 + 21hs^6s'^2u + 21hs^2s'^6u$$

$$- 14s^6s'^3 - 14s^3s'^6\Big\}/(30240h^2v^2 - 30240uh^2v) \quad (32)$$

For $hv \geq s > hu \geq s' > 0$:

$$\lim_{t_0 \to \infty} k_{|x_0,y_1,y_2,y_3}(t_0 + s, t_0 + s') =$$

$$\frac{s'^2}{30240h^2uv(u-v)}\Big[-21h^7u^6v^2 - 21hu^2s^6 - 21hs^2s'^4(u^2 - v^2) + 126h^2u^2vs^5$$

$$+ 126h^4u^4vs(h^2uv - hus - s^2) + 14s'(h^6u^6v + s^6u) + 14s^3s'^4(u-v) + 189h^5u^4v^2s^2$$

$$- 378h^5u^3v^3s^2 - 84h^4u^4vss'(hu - s) - 84huvs^5s' + 42h^4u^5s^2(s+s')$$

$$+ 42h^2ss'^4(u^2v - uv^2) + 252h^4u^2v^2s^2(us + vs + vs') - 168h^3uv^2s^2s'(hu^2 + us + vs)$$

$$- 315h^3u^2v^2s^4 - 56h^3u^4s^3s' + 112h^3u^3vs^3s' + 210h^4uv^4s4s' - 6h^2s'^5(u^2v - uv^2)\Big]$$

$$(33)$$

For $hv \geq s > hu > 0 \geq s'$:

$$\lim_{t_0 \to \infty} k_{|x_0, y_1, y_2, y_3}(t_0 + s, t_0 + s') =$$

$$\frac{s'^2}{4320 h^2 v(u-v)}\Big[ -3h^7 u^5 v^2 + 18h^6 su^4 v^2 + 2s'h^6 u^5 v - 18h^5 s^2 u^4 v + 27h^5 s^2 u^3 v^2$$

$$- 54h^5 s^2 u^2 v^3 - 12s'h^5 su^4 v + 6h^4 s^3 u^4 - 18h^4 s^3 u^3 v + 36h^4 s^3 u^2 v^2 + 36h^4 s^3 uv^3$$

$$+ 6s'h^4 s^2 u^4 + 12s'h^4 s^2 u^3 v - 24s'h^4 s^2 u^2 v^2 + 36s'h^4 s^2 uv^3 - 45h^3 s^4 uv^2 - 8s'h^3 s^3 u^3$$

$$+ 16s'h^3 s^3 u^2 v - 24s'h^3 s^3 uv^2 - 24s'h^3 s^3 v^3 + 18h^2 s^5 uv + 30s'h^2 s^4 s^2 - 3hs6u$$

$$- 12s'h^5 v + 2s's^6 \Big] \quad (34)$$

For $hv > hu \geq s, s' > 0$:

$$\lim_{t_0 \to \infty} k_{|x_0, y_1, y_2, y_3}(t_0 + s, t_0 + s') =$$

$$- \frac{s'^2}{30240 h^2 uv}\Big[ 126h^5 s^2 u^4 v - 378h^5 s^2 u^3 v^2 - 42h^4 s^3 u^4 + 84h^4 s^3 u^3 v + 252h^4 s^3 u^2 v^2$$

$$- 42h^4 s^2 s' u^4 + 84h^4 s^2 s' u^3 v + 252h^4 s^2 s' u^2 v^2 + 56h^3 s^3 s' u^3 - 336h^3 s^3 s' u^2 v$$

$$- 168h^3 s^3 s' uv^2 - 126h^2 s^5 uv + 210h^2 s^4 s' uv - 42h^2 ss'^4 uv + 6h^2 s'^5 uv$$

$$+ 21hs^6 v + 21hs^2 s'^4 u + 21hs^2 s'^4 v - 14s^6 s' - 14s^3 s'^4 \Big] \quad (35)$$

For $hv > hu \geq s > 0 \geq s'$:

$$\lim_{t_0 \to \infty} k_{|x_0, y_1, y_2, y_3}(t_0 + s, t_0 + s') =$$

$$- \frac{s^2 s'^2}{4320 h^2 uv}\Big[ 18h^5 u^4 v - 54h^5 u^3 v^2 - 6h^4 su^4$$

$$+ 12h^4 su^3 v + 36h^4 su^2 v^2 - 6s'h^4 u^4 + 12s'h^4 u^3 v + 36s'h^4 u^2 v^2 + 8s'h^3 su^3$$

$$- 48s'h^3 su^2 v - 24s'h^3 suv^2 - 18h^2 s^3 uv + 30s'h^2 s^2 uuv + 3hs^4 v - 2s's^4 \Big] \quad (36)$$

For $hv > hu > 0 \geq s, s'$:

$$\lim_{t_0 \to \infty} k_{|x_0, y_1, y_2, y_3}(t_0 + s, t_0 + s') =$$

$$\frac{hs^2 s'^2 u^2 \left( hsu - 4ss' + 3hs'u \right)}{2160v}$$

$$- \frac{s^2}{5040}\Big[ 21h^3 s'^2 u^3 - 63vh^3 s'^2 u^2 + 14h^2 ss'^2 u^2 + 42vh^2 ss'^2$$

$$+ 14h^2 s'^3 u^2 + 42vh^2 s'^3 u - 56hss'^3 u - 28vhss'^3$$

$$- s^5 + 7s^4 s' - 21s^3 s'^2 + 35s^2 + s'^3 \Big]$$

$$(37)$$

# D  Square-exponential kernel cannot yield Euler's method

We show that the square-exponetial (SE, aka. RBF, Gaussian) kernel cannot yield Euler's method for finite length-scales.

The SE kernel and its derivatives are

$$k(t, t') = \theta^2 \exp\left(-\|t - t'\|^2 / 2\lambda^2\right) \tag{38}$$

$$k^\partial(t, t') = \frac{(t - t')}{\lambda^2} k(t, t') \tag{39}$$

$$\partial k^\partial(t, t') = \left[\frac{1}{\lambda^2} - \left(\frac{(t - t')}{\lambda^2}\right)^2\right] k(t, t') \tag{40}$$

To show that this choice does not yield Euler's method, we proceed as in the case for the GMRK methods. The predictive mean after observing $x_0$ and $y_1$ is given by

$$\mu_{|x_0,y_1}(t_0 + s) = \begin{bmatrix} k(t_0 + s, t_0) & k^\partial(t_0 + s, t_0) \end{bmatrix} \underbrace{\begin{bmatrix} k(t_0, t_0) & k^\partial(t_0, t_0) \\ \partial k(t_0, t_0) & \partial k^\partial(t_0, t_0) \end{bmatrix}^{-1}}_{=:K} \begin{pmatrix} x_0 \\ y_1 \end{pmatrix}$$

$$= \begin{bmatrix} \theta^2 \exp(-\|s\|^2/2\lambda^2) & s\frac{\theta^2 \exp(-\|s\|^2/2\lambda^2)}{\lambda^2} \end{bmatrix} \begin{bmatrix} \theta^2 & 0 \\ 0 & \theta^2/\lambda^2 \end{bmatrix}^{-1} \begin{pmatrix} x_0 \\ y_1 \end{pmatrix}$$

$$= \begin{bmatrix} \exp(-\|s\|^2/2\lambda^2) & s \exp(-\|s\|^2/2\lambda^2) \end{bmatrix} \begin{pmatrix} x_0 \\ y_1 \end{pmatrix}$$

$$= \exp(-\|s\|^2/2\lambda^2)x_0 + s \exp(-\|s\|^2/2\lambda^2)y_1 \tag{41}$$

evaluated at $h$ yields

$$= \exp(-\|h\|^2/2\lambda^2)x_0 + h \exp(-\|h\|^2/2\lambda^2)y_1 \tag{42}$$

An interesting observation, left out in the main paper to avoid confusion, is that Eq. (42) *does* indeed produce the weights for Euler's method for the limit $\lim_{\lambda \to \infty}$. In fact, it can even be used to derive second and third order Runge-Kutta means, too. Future work will provide more insight into this property. However, this limit in the length-scale yields a Gaussian process posterior that has little use as a probabilistic numerical method, because its posterior covariance vanishes everywhere. This is in contrast to the integrated Wiener processes discussed in the paper, which yield proper finite, interpretable posterior variances, even after in the limit in $\tau$. Finally, SE kernel-GPs are not Markov. Inference in these models has cost cubic in the number of observations, reducing their utility as numerical methods.

## Footnotes

[1] http://probabilistic-numerics.org/ODEs.html