[Reviews · NeurIPS 2014]

Submitted by Assigned_Reviewer_5

The paper describes a probabilistic solver for ODE suing Runge-Kutta means. The idea is to find a Gauss-Markov process as probability distribution over the ODE solution. The posterior means match the outputs of Runge-Kutta family exactly, and the orders greater than 3 remains unidentified.

The paper is well-written. I am not qualified to judge contributions to the literature of ODEs. The idea to enable probabilistic numerical methods instead of point of estimate seems interesting. It would be helpful to bring in a real-world application to validate the value of the proposed method.
Summary: The paper is well-written. I am not qualified to judge contributions to the literature of ODEs.

Submitted by Assigned_Reviewer_9

1. Summary
This paper presents a probabilistic approach to ODE solvers based on Gaussian processes so that the posterior mean matches exactly the Runge-Kutta estimates. The main contribution of the paper is that it shows that Runge-Kutta solutions of order 1, 2, 3 can be obtained from integrated Weiner processes.

2. Quality
The paper is technically sound as it exploits the closeness property of GPs under linear operators to come up with covariance functions and, consequently, predictive means that match the Runge-Kutta solutions. The claims regarding the matching of these solutions are well supported by the analytical derivations in the paper and in the supplemental material. The claim regarding the low computational cost is also well supported, as it falls directly from the resulting model being a Gauss Markov process.

One possible weakness of this work is the claim in the main paper regarding previous ODE solvers that they do not match exactly the Runge-Kutta estimate. This contradicts what is presented in the supplementary material where the authors (line 448 onwards) that Equation (42) in the supplemental material can be used to derive 2nd order and 3rd order Runge-Kutta means. Although such a match is only achieved when \lambda tends to infinity, there should be more discussion in the main paper regarding both approaches. In fact, the authors do recognize (line 414) that the “application of different GP models might still be favorable in practice”, but fall short in providing more discussion or more experimentation comparing the proposed method with previous work.

One additional technical comment is the setting of the “scale” parameter \sigma^2, which seems to be done ad-hoc. Why haven’t he authors pursued a more principled way to provide point-estimates for this since it ultimately calibrates the the resulting posterior distribution?

3. Clarity
The paper is very well written and organized and it contains about the right balance of background and new material.

4. Originality
The paper addresses an existing problem, that of providing probabilistic interpretations for ODE solvers. It approaches this problem in a very different way to previous work as it goes beyond simply using GPs and exploiting their closeness under linear operators by showing equivalence results when starting from the kernel for the Wiener process.

5. Significance
The results are important as they provide a better understanding of Runge-Kutta methods from a probabilistic point of view. However, the paper will benefit greatly from a more careful comparison wrt to previous probabilistic approaches to the same problem.

6. Minor comments
(a) line 268: "Its finite values that this posterior indeed defines a proper GP". I don't think this sentence quite makes sense
(b) line 295: "an correction" --> "a correction"
Summary: A different perspective of probabilistic approaches to ODE solvers. A well written and theoretically sound paper that is well worth of being presented to the NIPS community. The paper will benefit greatly from a more careful theoretical and experimental comparison to previous work in the area.

Submitted by Assigned_Reviewer_16

The authors construct a family of Gaussian processes (GPs) whose predictive mean emulates an explicit Runge-Kutta solver for an initial value problem (IVP) exactly up to third order. In particular, they use integrated Wiener processes to obtain three covariance functions such that the GP prediction exactly equals the one step prediction of a Runge-Kutta scheme.

The work is interesting because it allows to approximate the solution of an ordinary differential equation (ODE) by a stochastic process in a principled way and hence provides a fresh look onto Runge-Kutta solvers.

The authors should more prominently state that their work covers explicit Runge-Kutta only and briefly name the differences between explicit and implicit methods.

I agree that Runge-Kutta methods have been studied for a long time and that there is no need to establish their approximation quality. However, there is always a remaining truncation error due to the limited number of elements from the Taylor series taken into account. There is a long literature on methods for estimating the (local) truncation error of (explicit) Runge-Kutta algorithms while computing the solution to the ODE/IVP. An empirical analysis and comparison of the exact truncation error, estimated truncation error and GMRK confidence band is the kind of information needed to pave the way to using the method in numerical integration practice.
Summary: The paper introduces GP covariance functions such that the mean prediction equals the output of explicit Runge-Kutta solvers up to third order. The construction enables a probabilistic treatment of the numerical solution process for initial value problems and suggests interesting future research directions.

Submitted by Assigned_Reviewer_41

Summary
Given an IPV, the paper constructs a family of probabalistic numerical methods that return a Gaussian-Markov process that defines a probability distribution over the IPV solution. In the construction the posterior means match the outputs of the Runge-Kutta family of first, second and third order. From a space of Gaussian process linear extrapolation methods with the property that the mean match the RK estimate a member is selected.

Quality
The quality of the paper is relativley good and comprehensive but unclear about how it will effect the performance on complex real problem. Related work seems to have been well cited but various details are missing which make it difficult to understand in certain locations.
example
p2 In Eq(3) without \sum_j w_{i,j}= c_i or explanation on b_i

Clarity
The paper is relatively clear but not sure how useful it maybe be in solving real problems. There seems to be some limitation but it is not stated clearly (eg p < 5).

Originality
Not the first attempt of finding an ODE solver that returns probability, but the proposed method provides certain renewed insights.

Significance
Application to a real problem would have been more signifcant.
Summary: This paper is signifcant in that it sheds light on the structure of the Runge-Kutta solver but not clear how this enlightment will be conducive in solving real problems.
Author Feedback
Author rebuttal: Many thanks to the reviewers for their valuable and positive feedback. We are happy to see the reviewers share our enthusiasm about this exact connection between GP regression and the established Runge-Kutta framework.

To address a general point:

=== Application to real-world problems ===

Because we are studying a connection between two very well-established frameworks, we opted to use the limited space to highlight big conceptual issues. A longer follow-up is in preparation, which will include extensive empirical evaluation on established benchmarks. The principal applied value of probabilistic ODE solvers as such has already been established by previous works, such as those by Hennig & Hauberg, and Schober et al., cited in the paper.

Individual comments:

=== Reviewer 16 ===

We will make the explicit-vs-implicit aspect more prominent than the current footnote on page 2.

=== Reviewer 41 ===

We will aim to clarify Equation 3 based on your comment. Regarding the limitation on order (p < 5), as noted in the paper, there is extensive and involved theory showing that the space of higher-order RK methods is intricately structured (e.g. the so-called "Butcher barrier" at p=5). This is carries over to GP formulations for higher order. "Leaving an issue for future work" is a much-abused excuse in rebuttals, but in this case there truly is a significant structural challenge in this direction, which does require
extensive mathematical footwork, way beyond the scope of this paper. A look at §2.5 in the textbook by Hairer & Wanner gives an idea of the complexity of this issue.

=== Reviewer 5 ===

Thanks for your review, see above for our argument on empirical applicability.

=== Reviewer 9 ===

Regarding the note in the supplements on SE-kernel GP models; this is a technical issue, left out of the main paper to avoid confusion: First, it turns out (this is not in this paper, as it requires further background) that higher-order (m-times) integrated Wiener kernels always also capture RK-rules of lower order (i.e. if m > p, the Theorems still hold). Since infinitely-integrated Wiener processes (infinite state-space models) are (nontrivially) connected to the SE kernel (see e.g. the discussion in Rasmussen & Williams, Appendix B and §4.2.1, and citations therein), it is not totally surprising that the SE can also be made to match RK. However, as noted in the supplements, this gives a pathological, and empirically useless model, because the match only appears in the limit of infinite length-scale. An SE-regressor with infinite length-scale can not meaningfully learn non-constant functions (much in contrast to the limit in our Wiener-formulation, which in some sense gives a "maximally flexible" Wiener model). We included this point in the supplements for completeness, but it has little applied value.

Regarding the choice of \sigma^2: The reviewer is correct that there are certainly several possible estimation rules for this parameter. But we disagree that our choice is ad-hoc, at least for a statistical estimator. It is based on a moment-matching of a particular projection of the inferred function, and leads to a well-scaled prior on the regressed function. It also has virtually no cost overhead. The reviewer is probably hinting at an inverse-Gamma prior as an alternative. This is indeed an interesting option, but note that it can not be trivially used in conjunction with noisy evaluations, which are an important application area for probabilistic solvers.